# SOLOS: Sparse Optimization for Long Sequences in Context Compression Enhanced LLMs

## Abstract

Recent advances in long-context large language models (LLMs) make them commercially viable, but their standard attention mechanisms' quadratic complexity hinders deployment due to excessive computational costs. To address this, researchers have explored Q-former-like architectures that compress input sequences for LLMs, reducing inference costs. However, these methods often underperform compared to mainstream LLMs trained on short sequences and struggle with longer context. We introduce SOLOS, an innovative method for training long sequences within limited computational resources. This approach effectively narrows the performance gap between context-compressed LLMs and mainstream LLMs handling long contexts. By significantly reducing training overhead, SOLOS enables training on long-sequence datasets, such as 100K tokens for instruction tuning, using merely an $8\times$ RTX3090 machine. Our comprehensive experimental analysis confirms SOLOS not only significantly outperforms other context-compression-augmented LLMs but also matches the performance of state-of-the-art long-context models. The introduction of SOLOS marks a significant step toward deploying long-context LLMs, offering both efficiency and effectiveness in practical scenarios.

## 1 Introduction

In recent years, long-context Large Language Models (LLMs) have seen rapid advancements, increasingly meeting commercial robustness standards (Li et al., 2023a; Team et al., 2024; Xinrong et al., 2024). Despite these advancements, deploying long-context LLMs in practical applications remains challenging, mainly due to the computational overhead from the standard full attention mechanism's quadratic complexity in long-context scenarios.

To address this issue, much research has focused on reducing computational burden. Notably, Wingate et al. (2022); Ge et al. (2024) found that LLM input tokens have considerable redundancy due to natural language's inherent redundancy. Building on this, Chevalier et al. (2023); Zhang et al. (2024) suggested compressing the input sequence by consolidating key information, reducing token count and computational costs. Many studies, including (Chevalier et al., 2023; Zhang et al., 2024), use a context encoder like BLIP-2's Q-former (Li et al., 2023b). This encoder integrates contextual information into learnable queries via an attention module. These queries, a new modality, must be aligned with the LLM's embedding space by a dedicated network before being input into the LLM.

Though these approaches offer significant acceleration through high compression rates, their performance often lags behind uncompressed models. The performance gap occurs because these models are typically trained on shorter sequences and directly applied to longer ones. For instance, Activation Beacon (Zhang et al., 2024) trains on up to 8K tokens but infers on sequences up to 400K tokens. The training-inference discrepancy limits the model's ability to handle long-context understanding. However, training these context-compression models on extremely long sequences is typically impractical due to excessive computational demands. Thus the challenge is reducing the training overhead for context-compression LLMs to effectively leverage long sequences.

We introduce **S**parse **O**ptimization for **LO**ng **S**equences (SOLOS), a context-compression framework with an efficient training methodology. Specifically, the context is divided into segments,

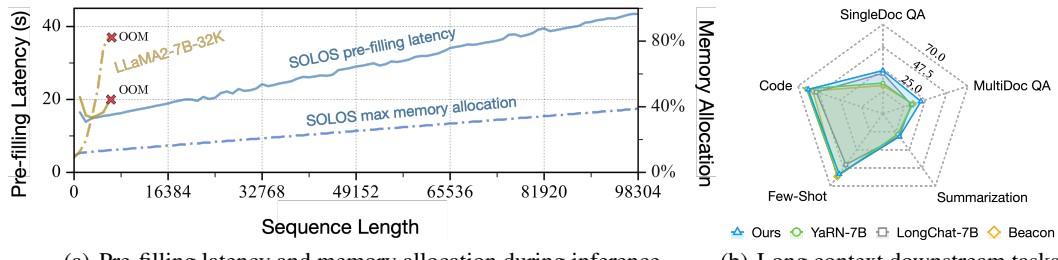

(a) Pre-filling latency and memory allocation during inference.    (b) Long context downstream tasks.

Figure 1: (a) End-to-end pre-filling latency and maximum GPU memory allocation comparison between SOLOS and LLaMA2-7B-32K (TogetherAI, 2023) across different sequence lengths. (b) Performance comparison between SOLOS and other long-context LLMs on downstream tasks.

each of which is appended with multiple special tokens at its end. After the forward propagation through the encoder, the activation of the special tokens have distilled contextual information, effectively forming a compact and informative condensed representation. This representation can be transferred to the decoder as additional key-value (KV) caches, via a projector consisting of two projection matrices. To minimize the introduction of additional parameters, we leverage LoRA (Hu et al., 2022) to fine-tune the encoder and the projector. This results in a mere 2% increase in parameters for LLaMA2-7B (Touvron et al., 2023). For optimization, we employ incremental computation exclusively on the decoder side. This means processing each segment sequentially, performing backpropagation immediately after each forward pass, and discarding activations to reduce memory usage—achieving up to an order-of-magnitude reduction. We avoid using incremental computation on the encoder side because it leads to extensive redundant computations. Instead, we use a reservoir sampling-based sparse optimization strategy to manage encoder activations, efficiently managing memory allocation without sacrificing long-term dependencies.

We conduct a comprehensive evaluation of SOLOS using LLaMA2-7B (Touvron et al., 2023) as our base model for a range of tasks. The tasks include auto-encoding, language modeling on datasets like PG19 (Rae et al., 2019), and long-context retrieval challenges like Needle In A Haystack (gkamradt, 2023). Additionally, we use the LongBench benchmark (Bai et al., 2023b) to assess SOLOS's real-world long-context performance. As illustrated in Figure 1, our findings show SOLOS achieves excellent compression at $8\times$ and $32\times$ ratios, allowing near-perfect reconstruction of the original context. Furthermore, SOLOS significantly outperforms other context-compression-enhanced LLMs across various tasks. Notably, SOLOS matches the performance of mainstream long-context LLMs in some evaluations, with significantly lower inference costs. This highlights SOLOS's potential to efficiently integrate long-context LLMs into practical scenarios.

## 2 RELATED WORKS

**Context-Compression-Enhanced LLMs.** For Large Multimodal Models, architectures like Q-former (Li et al., 2023b) have emerged as important methods for context compression, effectively condensing information from the context through a small number of learnable queries. Given the success of these techniques, they have gradually been integrated into the realm of LLMs. RMT (Bulatov et al., 2022) pioneers the application of this technique within language models, fusing information from each context segment iteratively into a fixed-size memory. This approach allows for efficient processing of long sequences at a minimal computational cost. Building on RMT, AutoCompressor (Chevalier et al., 2023) improves performance by concatenating representations from the context encoder across different segments. Activation Beacon (Zhang et al., 2024) expanded the trainable parameters and leveraged extensive instruction-tuning datasets, leading to even stronger performance on downstream tasks. However, because all these methods use the LLM as the context encoder, the training cost is prohibitively high, which limits the feasibility of training on long sequences. Consequently, the performance of these methods still falls short when compared to mainstream long-context models based on standard attention mechanisms.

**Long-Context LLMs.** Most mainstream long-context LLMs today are primarily pre-trained on short sequences; they then utilize position embedding extension techniques, coupled with limited

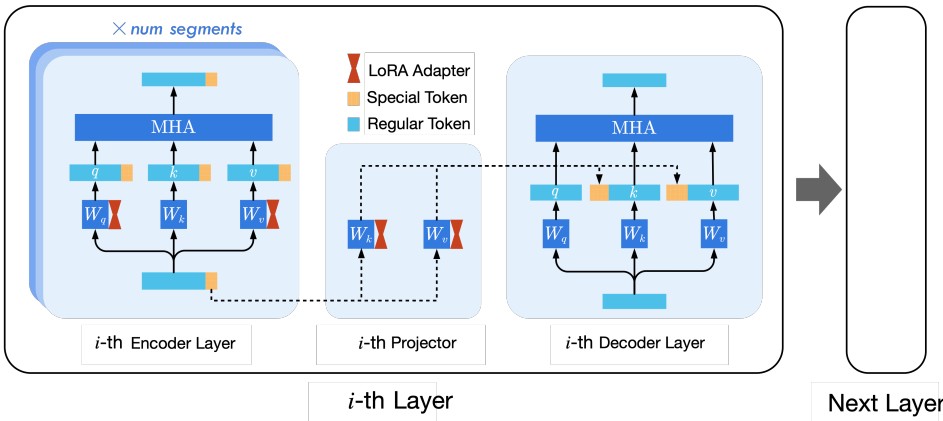

Figure 2: SOLOS uses an encoder-decoder structure. The context encoder's special tokens gather and blend information from regular tokens via attention. The combined representations are projected to the decoder. The encoder, decoder, and projector share parameters, but the encoder and projector have distinct LoRA adapters, treated as extra parameters, separate from model weights.

post-training on longer sequences, to handle extended contexts (Li et al., 2023a; TogetherAI, 2023). However, because these methods do not fundamentally alter the quadratic complexity of the standard attention mechanism, they continue to face prohibitive computational costs in real-world applications. For example, Huiqiang et al. (2024) reports that processing a 300K-token sequence with LLaMA3-8B (Dubey et al., 2024) requires 6 minutes on a single A100 machine, just to complete the pre-filling stage. This clearly demonstrates that excessive overhead hinders the commercial deployment of these long-context models. To address this issue, LongLoRA (Chen et al., 2023a) proposes using $S^2$ attention to replace standard full attention during training, significantly reducing the training overhead of long-context LLMs. However, since it still employs full attention during inference, the problem remains only partially resolved.

## 3 METHODOLOGY

### 3.1 OVERALL FRAMEWORK

The architecture of SOLOS, as depicted in Figure 2, is similar to Q-former. In this framework, each context segment is integrated into a compact assembly of special tokens. These special tokens, as context-rich representations, are projected into the decoder's embedding space for utilization.

### 3.2 STREAMLINED ENCODER-DECODER ARCHITECTURE

Our proposed model innovates by employing a decoder-only LLM as the context encoder, forming an encoder-decoder architecture. The encoder is akin to Q-former, but diverges from traditional methods (Chevalier et al., 2023; Zhang et al., 2024), where the encoder's output is typically liked directly to the decoder's initial layer. Instead, we introduce a parallel architecture. In this setup, the hidden states associated with special tokens from each encoder layer are mapped to the corresponding decoder layer to serve as compressed context. This mapping process is facilitated by a projector that transforms the hidden states into KV representations. These representations are then utilized directly as the KV cache for the decoder. To operationalize the projector, we harness the encoder's attention projection matrices $W_K$ and $W_V$ and incorporate LoRA adapters (Hu et al., 2022). This integration enhances the model's ability to adapt to new tasks with minimal additional parameters. Furthermore, to tap into the context encoder's potential for contextual summarization, we apply LoRA adapters to the encoder's $W_Q$ and $W_V$ projection matrices at each layer. These adapters are stored as separate parameters, rather than being integrated into the main model weights. This design choice allows for greater flexibility and control over the model's learning process.

Our architecture shares conceptual parallels with the Activation Beacon (Zhang et al., 2024), but with a novel twist: the deployment of two distinct sets of adapters. The first set is tasked with enhancing the projector's functionality, while the second is aimed at refining the encoder's capabilities. This dual-focus approach allows our model to more effectively capture and articulate the nuances of contextual information. We now present a formal mathematical description. Consider a given context $X$, which is partitioned into $k$ segments, each of length $l$, represented as $x_1, x_2, ..., x_k$. Additionally, there is a residual part $x_{k+1}$ that may be shorter than $l$.

**1) Pre-filling Stage.** *1.1) Special Token Appending:* In the initial layer of our model, we introduce special tokens of length $c$, denoted by $\pi$, to the end of each segment $x_1, x_2, ..., x_k$:

$$x_i \leftarrow x_i \oplus \pi, \quad \forall i \in [1, 2, 3, ..., k], \tag{1}$$

where $\oplus$ denotes the concatenation of the special tokens to the end of each segment.

*1.2) Hidden States Derivation:* In each layer's attention module, we derive the input hidden states $h_i$ corresponding to each segment $x_i$, where $h_i \in \mathbb{R}^{(l+c) \times d}$ and $d$ is the embedding dimension. *1.3) Special Token States Extraction:* Next, we extract the portion of the hidden states $h_i$ that corresponds to the special tokens, denoted as $h_\pi$. *1.4) Projection to KV Representation:* After passing $h_\pi$ through the projector, we obtain the key $K_i$ and value $V_i$ representations for each segment:

$$K_i, V_i \leftarrow \text{Projector}(h_\pi), \quad \forall i \in [1, 2, 3, .., k]. \tag{2}$$

*1.5) Concatentaion of KV Pairs:* The resulting key and value representations are concatenated to form the initial KV cache:

$$K_{\text{cache}} \leftarrow K_1 \oplus K_2 \oplus ... \oplus K_k, \quad V_{\text{cache}} \leftarrow V_1 \oplus V_2 \oplus ... \oplus V_k. \tag{3}$$

Finally, the concatenated $K_{\text{cache}}$ and $V_{\text{cache}}$ are fed into the decoder to be used as standard KV cache. The above process reflects the pre-filling stage.

**2) Decoding Stage and KV Cache Update.** During the decoding stage, new tokens are generated continuously. Once the total number of newly generated tokens, combined with the tokens from the residual part $x_{k+1}$, exceeds the size of one segment $l$, these tokens form a new segment. At this point, we can repeat the process of appending special tokens and projecting generate new $K_{k+1}$ and $V_{k+1}$. These new key-value pairs are then added to the existing KV cache:

$$K_{\text{cache}} \leftarrow K_{\text{cache}} \oplus K_{k+1}, \quad V_{\text{cache}} \leftarrow V_{\text{cache}} \oplus V_{k+1}. \tag{4}$$

### 3.3 Naive Optimization

**Segments Independence.** Though causal relationships exist among different segments, the compression process for each segment is independent and does not require the involvement of other segments. This allows segments to be processed separately, making a departure from previous approaches such as (Chevalier et al., 2023; Zhang et al., 2024), where later segments could leverage the fused representations from earlier ones.

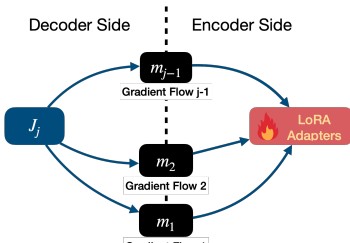

Figure 3: The gradients contains independent gradient flows.

**Gradient Expression.** The independent compression of each segment greatly simplifies the computational graph, enabling a concise expression for the parameter gradients. For the language modeling loss associated with the segment $x_j$, denoted as $J_j$, and considering the condensed representations $\{m_i\}_{i=1}^{i=j-1}$ of the $j-1$ preceding segments after passing through the encoder, the gradients can be efficiently calculated. These condensed representations are independent, acting as a relay during backpropagation to pass the gradient from the decoder to the trainable parameters in the encoder and the projectors, as depicted in Figure 3.

Assuming the LoRA adapters' trainable parameters are $\Theta$, the gradient for the $j$-th segment is:

$$\nabla_\Theta J_j = \sum_{i=1}^{j-1} \frac{\partial J_j}{\partial m_i} \cdot \frac{\partial m_i}{\partial \Theta}. \tag{5}$$

The final gradient is the cumulative sum of the gradients from all segments:

$$\nabla_\Theta = \sum_{j=1}^{k} \nabla_\Theta J_j. \tag{6}$$

**Challenge in Training on Long Sequences.** The gradient calculation as shown in Eq. (5) requires storing $k$ independent forward pass caches due to the independent generation of different $m_i$. Given that our encoder is based on an LLM, these caches are significantly large. For example, even with the use of gradient checkpoint, our test reveals that the memory capacity of an $8\times$ RTX3090 machine is limited to $k \leq 8$, posing a challenge for optimizing long sequences.

### 3.4 SPARSE OPTIMIZATION FOR LONG SEQUENCES

Running parallel forward and backward propagation for all segments from Eq. (6) can cause excessive GPU memory usage. To address this, we compute each segment one at a time and sum the results. This method, called incremental computation, balances the trade-off between time and memory, enabling training on longer sequences with limited resources.

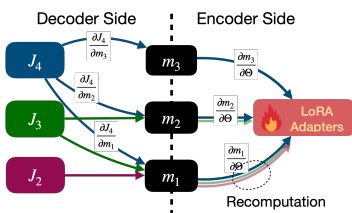

Figure 4: Recomputation.

**Incremental Computation Brings Recomputation.** Upon reviewing Eq. (5) and Eq. (6), we find that naive incremental computation leads to considerable recomputation, as shown in Figure 4. The derivative $\frac{\partial m_i}{\partial \Theta}$ is repeatedly calculated across segments, causing overhead and reducing its benefits.

**Incremental Computation on Decoder Only.** To reduce the excessive recomputation from the naive incremental method on the encoder, we focus incremental computation on the decoder. For each segment, we complete forward passes for both the encoder and decoder, but only apply backpropagation to the decoder. After processing all segments sequentially, $m_i$ sums the gradients from each segment's loss, as shown in the equation below:

$$\nabla_{m_i} = \sum_{j=i+1}^{k} \frac{\partial J_j}{\partial m_i}, \quad i \in [1, 2, ..., k]. \tag{7}$$

After accumulating gradients, we backpropagate through the encoder to determine the final gradients for $\Theta$. This approach, by conducting a single backpropagation through the encoder at the end, eliminates all unnecessary computations. The resulting final gradient matches that shown in Eq. (6). We will now demonstrate their equivalence.

*Proof.* We start by defining an indicator function $I(i, j)$ as:

$$I(i, j) = \begin{cases} 1 & 1 \leq j \leq k, \ 1 \leq i \leq j - 1, \\ 0 & \text{otherwise}, \end{cases} \tag{8}$$

where the condition can be inverted and explicitly solved as:

$$I(i, j) = \begin{cases} 1 & 1 \leq i \leq k, \ i+1 \leq j \leq k, \\ 0 & \text{otherwise}. \end{cases} \tag{9}$$

After applying this indicator function, we can rewrite Eq. (6) as follows:

$$\nabla_\Theta = \sum_{j=-\infty}^{+\infty} \left[ \sum_{i=-\infty}^{+\infty} I(i, j) \cdot \frac{\partial J_j}{\partial m_i} \cdot \frac{\partial m_i}{\partial \Theta} \right] = \sum_{i=-\infty}^{+\infty} \left[ \sum_{j=-\infty}^{+\infty} I(i, j) \cdot \frac{\partial J_j}{\partial m_i} \cdot \frac{\partial m_i}{\partial \Theta} \right],$$

$$= \sum_{i=1}^{k} \left[ \sum_{j=i+1}^{k} \frac{\partial J_j}{\partial m_i} \cdot \frac{\partial m_i}{\partial \Theta} \right] = \sum_{i=1}^{k} \left[ \left( \sum_{j=i+1}^{k} \frac{\partial J_j}{\partial m_i} \right) \cdot \frac{\partial m_i}{\partial \Theta} \right] = \sum_{i=1}^{k} \left( \nabla_{m_i} \cdot \frac{\partial m_i}{\partial \Theta} \right). \tag{10}$$

$\square$

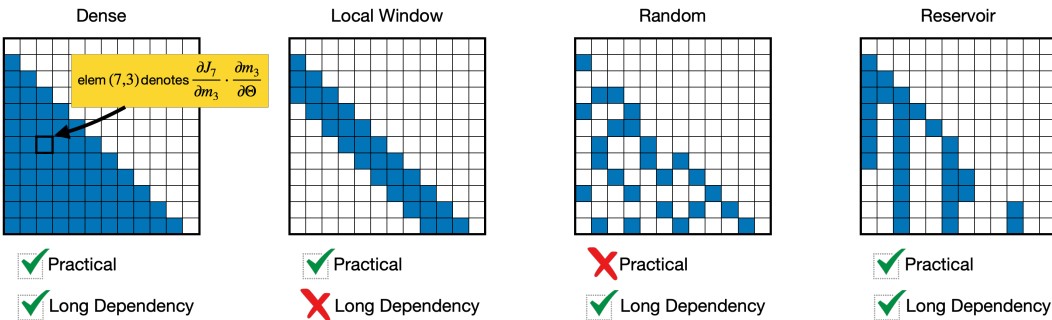

Figure 5: Comparison of different sparsity patterns. The sparsity achieved using reservoir sampling ensures both practicality and the ability to capture long-range dependencies.

**Sparse Optimization for Encoder Memory Reduction.** By applying incremental computation to the decoder, memory usage is stable since only the cache for one segment is needed, no matter the sequence length. This greatly reduces memory requirements, often leading to significant improvements. However, for the encoder, memory allocation remains high as all segment caches are stored, offering no reduction. Halting optimization here would only cut memory use in half without increasing FLOPs—a minor benefit, especially for very long sequences like 100K tokens. To cut encoder memory further, we suggest using a sparsity budget to limit the number of stored forward pass caches. if the cache limit is exceeded, we use an eviction strategy. For instance, if segment $i$ is removed, we first update $\nabla_\Theta$ by backpropagating $\nabla_{m_i}$, then apply a stop gradient to $m_i$ to halt further accumulation. We will now find the best eviction policy.

**Limitations of Local Window and Random Eviction Policies.** In Figure 5, the simplest eviction strategy is to keep only the most recent segment caches. This method ignores long-term dependencies and relies only on recent data for inference, reducing performance due to biased estimation. In contrast, random sparsity might ideally balance long-term and short-term dependencies, approaching the performance of dense optimization. However, since we cannot recover removed caches, we cannot randomly select segments from the complete set at each step.

**Reservoir Sampling for Eviction Policy.** To achieve efficient memory management, we need an eviction policy that integrates seamlessly with our incremental computation and provides unbiased gradient estimates. Reservoir Sampling (Vitter, 1985) meets both criteria. It offers a natural incremental processing mechanism that aligns with our sequential handling of segments. Additionally, its uniform sampling property ensures that the retained segments are a fair representation of the entire sequence, yielding unbiased gradient estimates of the true gradients. Next, we will demonstrate the expected gradient from reservoir sampling-based sparse optimization is equivalent to that of random sparse optimization.

*Proof.* We first express the uniform sampling property of reservoir sampling mathematically. Suppose we are processing the $j$-th segment ($j > S$). Let the binary random variables $\mathbf{Z}_{j,1}, \mathbf{Z}_{j,2}, \ldots, \mathbf{Z}_{j,j-1}$ denote the inclusion status of the previous $j - 1$ segments in the reservoir. For example, $\mathbf{Z}_{j,2} = 1$ indicates that the forward pass cache of the 2nd segment is retained in the reservoir, while $\mathbf{Z}_{j,1} = 0$ indicates that the forward pass cache of the 1st segment has been evicted. Since the size of our reservoir is fixed at $S$, the following constraint must hold:

$$\sum_{i=1}^{j-1} \mathbf{Z}_{j,i} = S. \tag{11}$$

Given these definitions, the uniform sampling property can be expressed as follows:

$$P(\mathbf{Z}_{j,1} = 1) = P(\mathbf{Z}_{j,2} = 1) = ... = P(\mathbf{Z}_{j,j-1} = 1) = \frac{S}{j-1}, \tag{12}$$

where $P(\triangle = 1)$ denotes the probability of the random variable $\triangle$ taking the value 1. In fact, random sampling also satisfies Eq. (12), but unlike reservoir sampling, it maintains uniform

---

**Algorithm 1** Sparse Optimization with budget size $S$

---

1: $R \leftarrow \emptyset$                 $\triangleright$ Initialize empty reservoir
2: **for** $i = 1$ to $T$ **do**
3:      $m_i \leftarrow \text{Encoder}(x_i)$       $\triangleright$ Perform forward pass through the encoder
4:      $J_i \leftarrow \text{Decoder}(x_i, \{m_j\}_{j=1}^{j=i-1})$       $\triangleright$ Perform forward pass through the decoder
5:      **backprop**$(J_i, \{m_j\}_{j \in R} \cup m_i)$       $\triangleright$ Backpropagate through the decoder only
6:      **if** $i < S$ **then**       $\triangleright$ If reservoir is not full, retain forward cache
7:          $R \leftarrow R \cup \{i\}$
8:      **else**       $\triangleright$ If reservoir is full, trigger eviction
9:          $j \leftarrow \text{randint}(1, i)$
10:         **if** $j < S$ **then**       $\triangleright$ Evict a previously saved segment $t$
11:             $t \leftarrow R[j]$
12:             $R[j] \leftarrow i$
13:             **stop_gradient**$(m_t)$
14:             **backprop**$(m_t, \Theta)$
15:         **else**       $\triangleright$ Or discard the incoming segment $i$
16:             **stop_gradient**$(m_i)$
17:             **backprop**$(m_i, \Theta)$
18:         **end if**
19:      **end if**
20: **end for**
21: **for** $i = 1$ to $S$ **do**       $\triangleright$ Backpropagate through the remaining segment
22:      $j \leftarrow R[i]$
23:      **backprop**$(m_j, \Theta)$
24: **end for**

---

sampling properties regardless of whether the previous step is observed or not:

$$P(\mathbf{Z}_{j,i}|\mathbf{Z}_{j-1,i} = 1) = P(\mathbf{Z}_{j,i}|\mathbf{Z}_{j-1,i} = 0), \quad \forall i \in [1, 2, ...., j-1]. \tag{13}$$

Nevertheless, without this property, reservoir sampling-based sparse optimization still achieves unbiased gradient estimation. Combining Eq. (5) and Eq. (6), we derive the expected final gradient:

$$
\begin{aligned}
\mathbb{E}_{\mathbf{Z}}[\nabla_\Theta] &= \mathbb{E}_{\mathbf{Z}}\left[ \sum_{j=1}^{k} \sum_{i=1}^{j-1} \sum_{z \in \{0,1\}} \left( z \cdot P(\mathbf{Z}_{j,i} = z) \cdot \frac{\partial J_j}{\partial m_i} \cdot \frac{\partial m_i}{\partial \Theta} \right) \right] \\
&= \mathbb{E}_{\mathbf{Z}}\left[ \sum_{j=1}^{k} \sum_{i=1}^{j-1} \left( P(\mathbf{Z}_{j,i} = 1) \cdot \frac{\partial J_j}{\partial m_i} \cdot \frac{\partial m_i}{\partial \Theta} \right) \right].
\end{aligned}
\tag{14}
$$

Here, the additional $\mathbf{Z}_{j,i}$ acts as a gate—if it is 0, it indicates that segment $i$ has been evicted when processing the $j$-th segment, thus making this term zero upon multiplication. Substituting Eq. (12) into Eq. (14) directly yields the expected value of the final gradient:

$$\mathbb{E}_{\mathbf{Z}}[\nabla_\Theta] = \sum_{j=1}^{k} \sum_{i=1}^{j-1} \left( \frac{S}{j-1} \cdot \frac{\partial J_j}{\partial m_i} \cdot \frac{\partial m_i}{\partial \Theta} \right) = \sum_{j=1}^{k} \frac{S}{j-1} \nabla_\Theta J_j. \tag{15}$$

We observe that this gradient is almost identical to the true gradient in Eq. (6), except for a factor of $S/(j-1)$. While this factor induces a systematic estimation error, it can be precisely offset by multiplying the resulting gradient $\nabla_\Theta J_j$ by a compensating factor, $(j-1)/S$, thereby enabling the reservoir sampling-based sparse optimization to achieve unbiased gradient estimation of Eq. (6).   $\square$

Using reservoir sampling-based sparse optimization on the encoder side maintains constant memory allocation regardless of sequence length, significantly reducing memory requirements while preserving gradient fidelity. We provide the detailed process in Algorithm 1 and further validate the unbiasedness of the gradient estimation in Appendix B.

## 4 EXPERIMENTATION

We evaluate SOLOS by: **(1) Lossless Compression of Contexts,** evaluated by an auto-encoding task introduced by (Ge et al., 2024). In this task, the input sequence undergoes a single forward pass through the context encoder to generate a compressed representation, which is subsequently utilized by the decoder to reconstruct the original input sequence. Superior performance is indicated by higher reconstruction fidelity. **(2) Long Context Language Modeling,** assessed using perplexity on the PG-19 (Rae et al., 2019), Proof-Pile (Azerbayev et al., 2022), and four distinct content categories from SlimPajama (Soboleva et al., 2023): Arxiv, Books, Github, and StackExchange. **(3) Retrieval,** evaluated through the Needle In A Haystack task (gkamradt, 2023), which scrutinizes the model's capacity to distill key information from arbitrary positions within the context. **(4) Long Context Downstream Tasks,** assessed on the LongBench (Bai et al., 2023b), encompassing a variety of subtasks: Single-Doc QA, Multi-Doc QA, Summarization, Few-shot, and Code. This suite of tasks provides a comprehensive evaluation of both comprehension and generative capabilities.

**Setups.** We use the LLaMA2-7B model (Touvron et al., 2023) with a 1K token segment size and compression ratios of 32 and 8. This configuration allows our model to support maximum context lengths of about 100K and 25K tokens, respectively. We apply LoRA fine-tuning (Hu et al., 2022) to both the encoder and the projectors, using a consistent configuration of $r = 32$ and $\alpha = 64$ for all LoRA modules. This allows us to adapt the pre-trained model to our specific task while maintaining a reasonable parameter count. Our training process has two stages. In the first stage, we train on 1B tokens from the SlimPajama dataset (Soboleva et al., 2023), building a strong foundation for language understanding. In the second stage, we fine-tune the decoder with a mixed dataset, comprising LongAlpaca (Chen et al., 2023b) (55.5%), Single-Detail QA (Zhang et al., 2024) (30%), BookSum (Kryściński et al., 2021) (12%), and Needle (gkamradt, 2023) (2.5%). We format these datasets into conversations using Vicuna's (Zheng et al., 2023) chat template, allowing our model to learn from diverse instructions and tasks. We use our proposed sparse optimization algorithm with a reservoir budget size of $S = 2$, enabling cache storage for up to 3 segments. We use the Adam optimizer at a 1e-4 learning rate with a cosine scheduler.

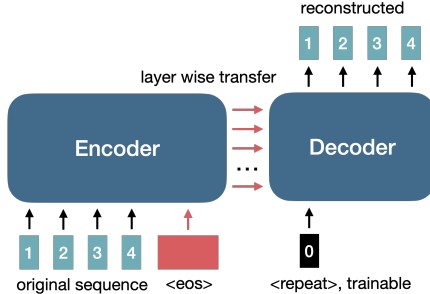

Figure 6: Auto-encoding task.

Table 1: Results of SOLOS and ICAE on the auto-encoding task.

|  | Ratio | BLEU-4↑ | Rouge-L↑ |
| --- | --- | --- | --- |
| SOLOS | 8 | 0.9851 | 0.993 |
|  | 32 | 0.5948 | 0.762 |
| ICAE | 8 | 0.6461 | 0.797 |
|  | 32 | 0.4289 | 0.585 |

**Auto-Encoding Task.** To evaluate information loss during context compression, we use the auto-encoding task introduced in (Ge et al., 2024), compressing and reconstructing text. We use the trained encoder to perform context compression. On the decoder side, we also train a "repeat" token as a signal for the auto-encoding task, as shown in Figure 6. We select the ICAE (Ge et al., 2024) as the baseline for comparing performance in reconstructing 1K-token sequences at 32× and 8× compression ratios. Table 1 shows the results. Our model performs well, achieving Rouge-L scores of 0.993 and 0.762 at at 32× and 8× compression ratios, respectively. To demonstrate our model's minimal information loss during compression, we randomly selected a sample and compared the original and reconstructed paragraphs, as shown in Figure 7. This significant reconstruction capability lays the foundation for using compressed context in inference.

**Long Sequence Language Modeling.** We assess SOLOS's long sequence language modeling using the PG19 (Rae et al., 2019) and ProofPile (Azerbayev et al., 2022) datasets, along with four SlimPajama (Soboleva et al., 2023) sub-datasets: Arxiv, Book, Github, and StackExchange, each with 100 randomly sampled instances. The results are summarized in Table 2. We compare SOLOS to baselines like LongChat-7B-v1.5-32K (Li et al., 2023a), LongAlpaca-7B-32K (Chen et al., 2023a), LLaMA2-7B-32K (TogetherAI, 2023), YaRN-7B-128K (Peng et al., 2024), and

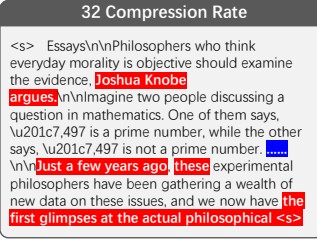

Figure 7: A case study of the auto-encoding task shows near-lossless compression at a ratio of 8. Even with a ratio of 32, reconstructed paragraphs retained meaning with minor wording changes.

Table 2: Results of various long-context LLMs on language modeling capability. "OOM" stands for Out-of-Memory error, which we've encountered upon an $8\times$ RTX3090 machine.

| Method | Ratio | PG19 | | | | | ProofPile | | | | | Arxiv | | Book | | Github | | StackExchange | |
|---|---|---|---|---|---|---|---|---|---|---|---|---|---|---|---|---|---|---|---|
| | | 4K | 16K | 25K | 32K | 100K | 4K | 16K | 25K | 32K | 100K | 25K | 100K | 25K | 100K | 25K | 100K | 25K | 100K |
| LongChat-7B | | 9.93 | 9.49 | 9.41 | 9.39 | | 5.65 | 3.90 | 3.56 | 3.21 | | 3.56 | | 6.91 | | 2.97 | | 8.77 | |
| LongAlpaca-7B | 1 | 9.96 | 9.75 | 9.69 | 9.67 | | 6.31 | 3.97 | 3.74 | 3.59 | | 3.71 | | 7.29 | | 3.09 | | 9.01 | |
| LLaMA2-7B-32K | | 7.06 | 7.17 | 7.15 | 7.14 | | **4.32** | **3.23** | **2.84** | **2.70** | | **2.85** | | 5.61 | | 2.37 | | **5.52** | |
| YaRN-7B-128K | | 6.54 | 6.62 | 6.60 | 6.58 | OOM | 4.45 | 3.32 | 2.93 | 2.79 | OOM | 3.07 | OOM | **5.43** | OOM | **2.36** | OOM | 5.67 | OOM |
| SOLOS | 8 | **6.27** | **6.09** | **6.09** | | | 4.43 | 3.84 | 3.41 | | | 3.32 | | 6.21 | | 2.46 | | 6.34 | |
| | 32 | 6.51 | 6.32 | 6.30 | **6.30** | **6.28** | 4.78 | 4.26 | 3.72 | 3.53 | **3.19** | 3.37 | **2.98** | 6.42 | **5.69** | 2.52 | **2.44** | 6.88 | **6.39** |
| Activation Beacon | 8 | 8.26 | 8.13 | 8.16 | | | 5.41 | 3.91 | 3.47 | | | 3.45 | | 6.67 | | 2.68 | | 8.33 | |
| | 32 | 8.56 | 8.54 | 8.58 | 8.59 | 8.83 | 5.79 | 4.33 | 3.86 | 3.70 | 3.33 | 3.51 | 3.78 | 6.93 | 7.92 | 2.76 | 3.05 | 8.45 | 9.91 |

Activation Beacon (Zhang et al., 2024). Our evaluation shows SOLOS performs comparably to non-compression models across sequence lengths and improves with length, unlike Activation Beacon. This is due to our protocol, which includes training on longer sequences, enabling SOLOS to capture long-range dependencies effectively and maintain performance on long sequences.

**Needle In A Haystack.** The Needle In A Haystack benchmark (gkamradt, 2023) assesses LLMs' ability to retrieve information from any context position. We assess both Activation Beacon and SOLOS, as shown in Figure 8. SOLOS performs flawlessly at $8\times$ compression and maintains a high pass rate even at $32\times$ compression. This indicates SOLOS can effectively use extended contexts, showing its remarkable efficiency in understanding long contexts.

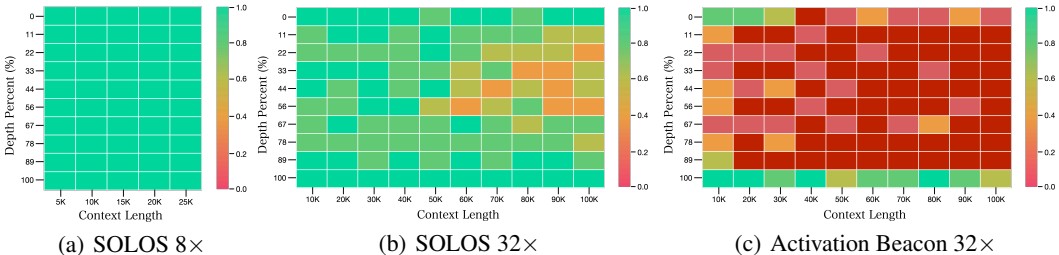

(a) SOLOS 8×      (b) SOLOS 32×      (c) Activation Beacon 32×

Figure 8: Results of SOLOS and Activation Beacon on the Needle In A Haystack test.

**LongBench.** We compare the performance of SOLOS on LongBench (Bai et al., 2023b) with LongChat-7B-v1.5 (Li et al., 2023a), LongAlpaca-7B (Chen et al., 2023a), Qwen1.5-7B-Chat (Bai et al., 2023a), Mistral-7B-Instruct-v0.2 (MistralAI, 2023), InternLM2-Chat-7B (Cai et al., 2024), GPT-3.5-Turbo-16K and Activation Beacon (Zhang et al., 2024). As shown in Table 3, SOLOS exhibits comparable performance to other LLaMA2-7B based models under both compression rates.

## 5 LIMITATIONS

Though SOLOS enhances training efficiency for sequences up to 100K tokens and performs well on downstream tasks, it faces limitations. One issue is SOLOS's reliance on a decoder-only LLM for context encoding instead of a specialized, newly trained encoder. This can lead to significant data loss at higher compression levels, limiting efficiency improvements. Moreover, SOLOS's use of an

Table 3: Results of various long-context LLMs on five subtasks from LongBench. † denotes results from the LongBench paper.

| Model | Ratio | SQA | MQA | SUM | FEW | CODE |
|---|---|---|---|---|---|---|
| *LLaMA2-7B / LLaMA2-7B-chat based* | | | | | | |
| LongChat-7B-32K | | 31.6 | 23.5 | 21.7 | 49.3 | 54.9 |
| LongAlpaca-7B-16K | 1 | 26.6 | 28.0 | 24.5 | 52.9 | 52.4 |
| YaRN-7B-128K | | 24.0 | 24.1 | 19.8 | 60.0 | 62.71 |
| Activation Beacon | 8 | 22.1 | 24.8 | 20.2 | 60.8 | 57.7 |
| | 32 | 19.8 | 23.4 | 18.0 | 58.3 | 56.2 |
| SOLOS | 8 | 33.8 | 31.3 | 22.1 | 58.3 | 61.5 |
| | 32 | 28.5 | 26.9 | 20.3 | 57.0 | 60.8 |
| *Others* | | | | | | |
| Qwen1.5-7B-Chat | | 27.9 | 14.2 | 21.0 | 21.8 | 28.9 |
| Mistral-7B-Instruct-v0.2 | | 31.3 | 26.4 | 21.8 | 46.6 | 44.8 |
| †GPT-3.5-Turbo-16K | 1 | 45.1 | 36.2 | 23.9 | 52.9 | 54.1 |
| InternLM2-Chat-7B | | 45.7 | 43.1 | 26.5 | 58.3 | 36.4 |

LLM for context encoding is computationally intensive. Employing simpler models might compress context more efficiently, reducing costs and enhancing performance. Future research could benefit from investigating more efficient encoding techniques. Finally, SOLOS requires additional training, potentially making it less convenient than methods that bypass training requirements.

## 6 CONCLUSION

To enhance the ability of LLMs on processing long sequences, We have proposed SOLOS, which employs a streamlined encoder-decoder framework where the weights-shared encoder and decoder respectively encapsulate a context segment into compressed representations and leverage these representations to predict outputs of the subsequent segment. Moreover, we introduce two strategies for reducing memory allocation in the encoder and decoder. For the decoder, we adopt incremental computation, which processes segments sequentially rather than in parallel, significantly reducing memory footprint without increasing FLOPs. For the encoder, we apply reservoir sampling-based sparse optimization, an unbiased method that balances efficiency and gradient accuracy. With these optimizations, SOLOS can be efficiently trained on sequences of 100K tokens with limited resources, resulting in a strong performance on language modeling tasks and comparable performance on various downstream tasks.

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

## A  IMPLEMENTATION DETAILS

**Overview of Pretraining Data.** During the pretraining phase, we use a total of 1 billion tokens from five sub-datasets of SlimPajama. For Github, StackExchange, and Wiki, most of the data consists of short sequences, while for Book and Arxiv, the majority of the data consists of long sequences. We randomly sample tokens from each dataset for training, with a total of 1 billion tokens sampled across all datasets. Detailed information is provided in Table 4.

Table 4: Detailed information on pretraining data.

| Corpus | Num Seq. | Num Sample Token | Sequence Length | | |
|---|---|---|---|---|---|
| | | | Max | Min | Average |
| Book | 13K | 0.15B | 18M | 29 | 510K |
| Arxiv | 101K | 0.15B | 3M | 217 | 57K |
| Github | 1M | 0.4B | 1M | 571 | 6K |
| StackExchange | 1.3M | 0.15B | 267K | 1K | 3K |
| Wiki | 1.3M | 0.15B | 452K | 523 | 3.5K |

**Overview of Instruction Tuning Data.** In instruction tuning, we add LoRA adapters (Hu et al., 2022) to the query and value projection matrices of each layer in the decoder. We organize all instruction tuning data into a conversation format following Vicuna's chat template (Zheng et al., 2023), as shown below: "A chat between a curious user and an artificial intelligence assistant. The assistant gives helpful, detailed, and polite answers to the user's questions. ### USER: ⟨Request⟩ ### Assistant: ⟨Response⟩" We finetune on multiple instruction tuning datasets, most of which are using ChatGPT for response generation.

# B  ADDITIONAL EXPERIMENTATION

**Ablation on LoRA Hyperparameters.** During pretraining, we add LoRA adapters to the query and value projection matrices of all encoder layers, as well as to the key and value projection matrices of all projectors. In the instruction tuning phase, in addition to the LoRA adapters used in pretraining, we also add LoRA adapters to the query and value projection matrices of each decoder layer. To assess the impact of different LoRA configurations on the final results, we train with a $32\times$ compression ratio using the same training recipe and evaluated the resulting models on PG19 and SingleDoc QA from LongBench. The results are shown in Table 5. The experimental results indicate that larger rank and alpha values improve language modeling performance but negatively affect instruction-following capabilities.

Table 5: Performance comparison under different LoRA configurations.

| LoRA Rank | LoRA Alpha | PG19 | | | | SQA |
|---|---|---|---|---|---|---|
| | | 4K | 16K | 32K | 100K | |
| 32 | 64 | 6.51 | 6.32 | 6.30 | 6.28 | **28.5** |
| 64 | 128 | 6.43 | 6.23 | 6.21 | 6.20 | 27.7 |
| 128 | 256 | **6.41** | **6.22** | **6.20** | **6.19** | 27.3 |

**Ablation on Sparse Optimization Algorithm.** Local window sparse optimization only considers the most recent segments, theoretically favoring local dependencies while overlooking long-term ones. This prevents the full utilization of long context, making it a suboptimal approach. To evaluate the actual performance of local window sparse optimization and quantify the improvements brought by SOLOS, we use the same training recipe to compare the models trained with these two optimization algorithms. We assess their language modeling performance on PG19 and their performance on the SingleDoc QA task from LongBench, as shown in Table 6. The results demonstrate that models trained using local window sparse optimization fail to capture long-term dependencies, causing language modeling perplexity to stagnate as context length increases. In contrast, SOLOS addresses this limitation effectively.

Table 6: Performance Comparison between local window sparse optimization and SOLOS

| Local Window Sparse | SOLOS | PG19 | | | | SQA |
|---|---|---|---|---|---|---|
| | | 4K | 16K | 32K | 100K | |
| ✓ | | **6.42** | 6.45 | 6.57 | 6.51 | 22.1 |
| | ✓ | 6.51 | **6.32** | **6.30** | **6.28** | **28.5** |

**Accuracy of Gradient Estimation.** To assess the accuracy of SOLOS in gradient estimation, we compare its gradients with those from Eq. (6). We expect high similarity, indicating our method's accuracy. We use a context window of 128, compression ratio of 8, and compute gradients for parameters $\Theta$ based on a mini-batch of 64 inputs, each with 2048 tokens. To avoid the influence of initial trainable parameter values, we use the same checkpoint for both approach after hundreds of training iterations. We use $\nabla_\Theta^*$ to represent the gradient from our reservoir sampling-based sparse optimization and compute the similarity ratio $r = \|\nabla_\Theta^*\| / \|\nabla_\Theta\|$. A ratio of $r = 1$ indicates the gradients are close. Table 7 shows the mean and variance of $r$ for different reservoir budget. Our results demonstrate that applying the compensating factor $\frac{i-1}{S}$ ensures high similarity between the estimated and true gradients, even when the reservoir budget is as low as 1.

Table 7: A statistical analysis compares the L2 norm ratios of gradients from two algorithms. Key findings: 1) The compensating factor is vital for accurate gradient estimation. 2) Variance decreases with increasing window size $S$, enhancing estimation accuracy.

| Statistics | w/o Factor | | | | w/ Factor | | | |
|---|---|---|---|---|---|---|---|---|
| | $S=1$ | $S=4$ | $S=8$ | $S=\infty$ | $S=1$ | $S=4$ | $S=8$ | $S=\infty$ |
| Mean | 0.676 | 0.818 | 0.935 | 1.000 | 0.994 | 0.980 | 1.024 | 0.999 |
| Variance | 0.112 | 0.041 | 0.008 | 3e-5 | 0.039 | 0.038 | 0.009 | 4e-5 |

