# OpenReview forum: "SOLOS: Sparse Optimization For Long Sequence In Context Compression Enhanced LLMs"
_ICLR.cc/2025/Conference — Submitted to ICLR 2025_

### Official Review · Reviewer_dvE7 · 2024-10-29

**Soundness:** 2
**Presentation:** 2
**Contribution:** 2
**Rating:** 5
**Confidence:** 3

**Summary:**

This paper introduces SOLOS, a method aimed at reducing resource consumption in long-context scenarios. The approach utilizes an intra-layer parallel encoder-decoder structure to compress context and employs sparse optimization to reduce memory usage during training.

**Strengths:**

By compressing the context, the proposed approach lowers memory usage and improves generation speed for long-context models.

1. Efficient Architecture

The model compresses context at a fixed rate, utilizing an encoder-decoder structure within each layer. The encoder compresses fixed-length segments into shorter special tokens, with shared encoder-decoder parameters adapted using LoRA.

2. Memory-Efficient Training

The training process maintains fixed memory usage by applying recomputation and sparse optimization based on Reservoir Sampling.

**Weaknesses:**

The primary issue with this paper is the lack of experimental coverage.

1. The experiments only compare the model with ICAE on auto-encoding tasks, lacking comparisons on other tasks. The Activation Beacon is also not evaluated on auto-encoding tasks.

2. Other key-value (KV) cache compression methods, besides the Q-former-like approach, need to be considered in comparisons.

3. Comparison of memory and speed with other methods is absent; only full-attention is compared, without evaluating other context compression techniques.

Minor Issues:

Citation for the baseline method Activation Beacon appears to be incorrect.

**Questions:**

Why is LoRA not applied to $W_k$ in the encoder?

---

### Official Review · Reviewer_R53E · 2024-11-01

**Soundness:** 3
**Presentation:** 3
**Contribution:** 2
**Rating:** 5
**Confidence:** 4

**Summary:**

This papper proposes SOLOS, which uses limited computational resources  for efficiently training long-context LLMs. Standard attention mechanisms in LLMs have quadratic computational complexity, making them computationally expensive for long sequences. Previous approaches, like Q-former architectures that compress input sequences, reduce inference costs but often underperform with longer contexts compared to mainstream LLMs. SOLOS addresses these challenges by significantly reducing training overhead, enabling the training of long-sequence datasets—up to 100,000 tokens—for instruction tuning.

**Strengths:**

The problem focused in this paper is very important

**Weaknesses:**

First of all, many important long contex tasks are very sensitive to token eviction based compression. For example:

Given a long piece of material and then engages in a multi-turn conversation about it. Since questions are asked sequentially and depend on previous answers, it's impossible to predict which parts of the material will be important for all possible questions at the beginning. Unfortunately, multi-turn conversations are the most common use case for chatbots.

Second, the reservoir pattern in this paper is very similar to the heavy hitter in H2O [1]. However, there is no discussion on this.

Third, distilling the context into special tokens have been already proposed in LongT5 [2]. There is no discussion or compairison.

Also, reservoir pattern is hard to be compatible with Flash Attention, which is a must-use in long context scenario. Can the author explain how you implement it in practice?

Finally, please clearly state your setting of Needle test. Needle test is very sensitive to the prompt given to the model.


[1] H2O: Heavy-Hitter Oracle for Efficient Generative Inference of Large Language Models

[2] LongT5: Efficient Text-To-Text Transformer for Long Sequences

**Questions:**

See Weaknesses

---

### Official Review · Reviewer_gxfu · 2024-11-02

**Soundness:** 3
**Presentation:** 3
**Contribution:** 3
**Rating:** 6
**Confidence:** 3

**Summary:**

The paper proposes SOLOS, a novel approach to training long-context LLMs and addresses the computational challenges of handling long sequences. The method uses a context compression framework with an encoder-decoder architecture, where the context is divided into segments and appended with special tokens. The key innovation lies in its efficient training methodology that combines incremental computation on the decoder side with reservoir sampling-based sparse optimization for the encoder, allowing training on sequences up to 100K tokens using modest computational resources (8× RTX3090).

**Strengths:**

1. The approach tackles an important real-world problem, the computational overhead of deploying long-context LLMs.
2. The combination of incremental computation and reservoir sampling is novel to me.
3. SOLOS matches or exceeds the performance of mainstream long-context LLMs while requiring significantly less computational resources.

**Weaknesses:**

1. Although SOLOS extends the context length to 100K with a ratio of 32, its perplexity often increases compared to using a ratio of 8. This suggests SOLOS may not fully leverage the 100K context length.
2. SOLOS appears to be significantly slower than regular LLMs when operating under a 4K context length.
3. The evaluation mainly focuses on LLaMA2-7B as the base model, which raises questions about its generalizability to other configurations, such as larger or more recent models.

**Questions:**

1. Have you tried using RULER to validate the context length of the trained model under different ratios?

---

### Official Review · Reviewer_o2QR · 2024-11-12

**Soundness:** 3
**Presentation:** 2
**Contribution:** 2
**Rating:** 3
**Confidence:** 3

**Summary:**

This paper presents a method that first chunks input into segments and then attaches special tokens at each segment. There are two LLMs at play, with the first one taking the activation of the special tokens, going through a projector, and becoming some additional KV cache in the second LLM, which is responsible for the actual output generation. The first LLM and the projector are LoRA-tuned.

**Strengths:**

Advancing prefill compression techniques is a vital part of LLM efficiency, as long-context problems often come in this manner.

Performance demoed in Figure 7 is very impressive.

**Weaknesses:**

1. The idea of chunk processing a long input and using some special tokens per chunk to register context information is well explored. We have activation beacon (as mentioned by the authors), gist [1], ultragist [2] (both are not mentioned.), and maybe many other methods I don't know about have gone through this route. This limits the novelty of this work.

2. Following #1, I am not sure if the more complicated design of the proposed method (2 LLMs in parallel, a projector, etc.) is worthwhile. There is virtually no efficiency comparison outside Figure 1(a), which only compares against the vanilla full attention baseline.

3. The performance evaluation of this paper really only features 1 comparative method (activation beacon), as all other baselines are long context-tuned LLMs with full attention. This coverage is too thin.

4. Following #3. The evaluation also really only applies the proposed method to one model (llama2-7b) and is mainly tested on one real long context dataset (longbench), which is, again, thin on coverage and a very dated model choice. I am interested in discussion/comparison wrt gist/ultragist, TOVA [3], DMC [4], and maybe even inference-only sparsity methods like MInference [5] and SnapKV [6]. As well as results on Rulers and InfiniteBench.

5. The position of the paper is a bit vague. The experiment layout of this paper is akin to a long-context compression method that requires finetuning, but the paper heavily focuses on how it enables better long-context training. However, it looks like the support for the training claim comes down to being able to compress prefill and using LoRA for weight update. Which are 1) not something unique to the proposed method and 2) vastly underexplored compared to works studying efficient long-context training, such as [7].

6. The paper has significant delivery issues, mostly revolving around having a rather shallow capture of existing works. For example, the authors mentioned their architectural similarity with activation beacon around line 162, but without diving into any details. The authors introduced reservoir sampling (a general randomized sampling technique) around line 300, but again did not provide much information regarding its adaptation in LLM nor how it solves the "cache recover" problem of pure random sampling (which is also an under-explained challenge).

Overall, I think the method has good potential, mainly because of the performance demonstrated in Figure 7. It is rare to see methods doing compression at the token level able to recover the original text in natural language this well. However, I am afraid the delivery and evaluation coverage of the current work require much improvement.

[1] Learning to Compress Prompts with Gist Tokens
[2] Compressing Lengthy Context With UltraGist
[3] Transformers are Multi-State RNNs
[4] Dynamic Memory Compression
[5] MInference 1.0: Accelerating Pre-filling for Long-Context LLMs via Dynamic Sparse Attention
[6] SnapKV: LLM Knows What You are Looking for Before Generation
[7] How to Train Long-Context Language Models (Effectively)

---

Update: I recently came to realize that ultragist is the same as activation beacon per https://github.com/namespace-Pt/UltraGist/issues/4. Given the authors have already compared with activation beacon, please ignore my requests regarding ultragist above. Sorry for the overlook.

**Questions:**

1. What's the background filler for your needle task? If it is the reputation of the same task, please consider rerunning it with a noisy background (or just do a full Ruler).

2. Did you truncate the longbench input for different models/baselines?

---

### Meta-Review · Area_Chair_G3uP · 2024-12-23

**Metareview:**

SOLOS presents a method for training long-sequence LLMs with limited computational resources, enabling training on sequences up to 100K tokens using an 8x RTX3090 machine. The paper's strengths lie in addressing a critical practical problem in LLM deployment, demonstrating efficiency gains, and showing promising performance in specific metrics. However, significant weaknesses include limited novelty (with core ideas appearing in existing work like activation beacon and gist), insufficient experimental validation (primarily testing on LLaMA2-7B with few comparative baselines), technical concerns (including Flash Attention compatibility issues and unclear benefits of the complex architecture), and presentation issues (shallow treatment of related work and unclear positioning). The paper received low scores (3, 5, 5, 6) placing it in the bottom quartile of submissions. These factors, combined with multiple unaddressed technical concerns and unclear justification for architectural complexity, lead to a reject recommendation. The decision is particularly supported by the insufficient experimental validation and limited novelty relative to existing methods.

**Additional Comments On Reviewer Discussion:**

Regarding the rebuttal period, notably there appears to be no author response or rebuttal discussion in the provided review thread, despite significant concerns raised by all reviewers. Reviewer o2QR questioned novelty and requested additional baseline comparisons, Reviewer R53E raised concerns about token eviction in multi-turn conversations and similarities to H2O, Reviewer dvE7 highlighted limited experimental coverage, and Reviewer gxfu questioned performance degradation at higher compression ratios. The absence of author responses to these substantial technical questions and requests for clarification is concerning and weighs negatively in the final decision. This lack of engagement during the rebuttal period, combined with the relatively low scores and consistent concerns across all reviews about novelty, experimental validation, and technical clarity, strongly supports the reject recommendation as these important issues remain unaddressed.

---

### Decision · Program_Chairs · 2025-01-22

Reject